# Integrated Immunohistochemical Study on Small-Cell Carcinoma of the Lung Focusing on Transcription and Co-Transcription Factors

**DOI:** 10.3390/diagnostics10110949

**Published:** 2020-11-13

**Authors:** Younosuke Sato, Isamu Okamoto, Hiroki Kameyama, Shinji Kudoh, Haruki Saito, Mune Sanada, Noritaka Kudo, Joeji Wakimoto, Kosuke Fujino, Yuki Ikematsu, Kentaro Tanaka, Ayako Nishikawa, Ryo Sakaguchi, Takaaki Ito

**Affiliations:** 1Department of Pathology and Experimental Medicine, Graduate School of Medical Sciences, Kumamoto University, Kumamoto 860-8556, Japan; nosuke2014@gmail.com (Y.S.); kudoh@kumamoto-u.ac.jp (S.K.); h-saito@kumamoto-ent.com (H.S.); msanadakuma@icloud.com (M.S.); iha.jinjutsu@gmail.com (N.K.); literaryflower@gmail.com (A.N.); sakaryo.soccer@icloud.com (R.S.); 2Research Institute for Diseases of the Chest, Graduate School of Medical Sciences, Kyushu University, Fukuoka 812-8582, Japan; okamotoi@kokyu.med.kyushu-u.ac.jp (I.O.); yuki-ike@kokyu.med.kyushu-u.ac.jp (Y.I.); tanaka-k@kokyu.med.kyushu-u.ac.jp (K.T.); 3Department of Medical Examination, Faculty of Health Sciences, Kumamoto Health Science University, Kumamoto 861-5598, Japan; hirokame@kumamoto-hsu.ac.jp; 4Department of Thoracic Surgery, Graduate School of Medical Sciences, Kumamoto University, Kumamoto 860-8556, Japan; 5Department of Otolaryngology-Head and Neck Surgery, Graduate School of Medical Sciences, Kumamoto University, Kumamoto 860-8556, Japan; kfujino@kumamoto-u.ac.jp; 6National Hospital Organization Minami-Kyushu National Hospital, Kagoshima 899-5293, Japan; wakimojo@skyusyu2.hosp.go.jp

**Keywords:** small-cell lung cancer, immunohistochemistry, ASCL1, NEUROD1, YAP1, POU2F3

## Abstract

Small-cell lung cancer (SCLC) is an aggressive malignant cancer that is classified into four subtypes based on the expression of the following key transcription and co-transcription factors: ASCL1, NEUROD1, YAP1, and POU2F3. The protein expression levels of these key molecules may be important for the formation of SCLC characteristics in a molecular subtype-specific manner. We expect that immunohistochemistry (IHC) of these molecules may facilitate the diagnosis of the specific SCLC molecular subtype and aid in the appropriate selection of individualized treatments. We attempted IHC of the four key factors and 26 candidate SCLC target molecules selected from the gene expression omnibus datasets of 47 SCLC samples, which were grouped based on positive or negative results for the four key molecules. We examined differences in the expression levels of the candidate targets and key molecules. ASCL1 showed the highest positive rate in SCLC samples, and significant differences were observed in the expression levels of some target molecules between the ASCL1-positive and ASCL1-negative groups. Furthermore, the four key molecules were coordinately and simultaneously expressed in SCLC cells. An IHC study of ASCL1-positive samples showed many candidate SCLC target molecules, and IHC could become an essential method for determining SCLC molecular subtypes.

## 1. Introduction

Neuroendocrine tumors account for approximately 20% of all lung cancers, and small-cell lung cancer (SCLC) accounts for approximately 70% of all neuroendocrine tumors [1,2]. SCLC exhibits clinically aggressive malignant behavior. It is mainly a high-grade neuroendocrine cancer that is characterized by rapid tumor growth, high vascularity, early metastasis, high sensitivity to radiotherapy and chemotherapy, multidrug resistance, and the inactivation of TP53 and RB1 [3,4,5]. The main risk factor for SCLC is smoking as more than 95% of SCLC patients are current or ex-smokers [3]. Surgical therapy for SCLC is not generally performed in the majority of patients diagnosed with SCLC because of its rapid metastasis to the lymph nodes, the other lung, and organs; therefore, the first-line treatment is mainly chemotherapy with or without radiation [6,7]. However, standard chemotherapies for SCLC have been performed for a few decades, and these therapies have not shown satisfactory effects on the prognosis of SCLC patients; therefore, it is necessary to identify and develop novel therapeutic approaches to treat SCLC [8,9]. Recent clinical studies on the anti-PDL1 antibody atezolizumab for advanced SCLC reported no significant effects on patient prognosis [10]. To develop more effective molecular targeted therapies, Rudin et al. recently proposed the classification of SCLC into four molecular subtypes based on the expression of the following key transcription and co-transcription factors: achaete-scute homologue 1 (ASCL1), neurogenic differentiation factor 1 (NEUROD1), yes-associated protein 1 (YAP1), and POU class 2 homeobox 3 (POU2F3) [11].

The use of immunohistochemistry (IHC) to diagnose neuroendocrine tumors, including SCLC, is not crucial under the current WHO classification of lung cancer. The examination of hematoxylin and eosin-stained slides is the primary method used to make diagnoses [12]. However, IHC can discriminate between neuroendocrine and non-neuroendocrine tumors to determine cytological and tissue diagnoses, for which the neuroendocrine markers synaptophysin, chromogranin A, and NCAM1, also known as CD56, are very useful [13]. Recent studies have identified insulinoma-associated protein 1 (INSM1) as a sensitive and specific neuroendocrine marker for the diagnosis of all neuroendocrine tumors including SCLC [14,15]. Ki67 and thyroid transcription factor 1 (TTF1) are also useful for the differential diagnosis of SCLC [13]. Regarding the relationship between SCLC and neuroendocrine features, small undifferentiated tumors in the lungs were originally referred to as oat-cell carcinoma in the 1920s [16]. In the 1960s, Bensch et al. used electron microscopy to demonstrate that SCLC contains neurosecretory-type granules in the cell cytoplasm [17]. Additionally, the development of IHC and the identification of neuroendocrine markers provided support for the relationship between neuroendocrine features and SCLC. IHC for SCLC using the neuroendocrine markers chromogranin A, NCAM1, and INSM1 is regarded as an adjunctive tool.

SCLC was previously considered to be a “homogenous” cancer because of the highly frequent inactivation of TP53 and RB1 [18,19]. However, SCLC had also been considered a “heterogeneous” cancer with neuroendocrine features and morphological characteristics. Based on these characteristics, SCLC cell lines are classified into two types: a “classic” subtype with typical morphology and neuroendocrine-like features and a “variant” subtype with an atypical morphology [20,21]. Poirier et al. reported that the classic and variant subtypes correlate with the expression of the lineage transcription factors ASCL1 and NEUROD1 [22]. ASCL1 is a neuroendocrine lineage master regulator of the lungs, and NEUROD1 also contributes to the regulation of neuroendocrine cell development in the lungs [22,23]. In contrast to classic SCLC, which strongly expresses ASCL1, variant SCLC strongly expresses NEUROD1 [18]. The non-neuroendocrine subtype weakly expresses both ASCL1 and NEUROD1, while YAP1, a co-transcription factor activated by the HIPPO signal pathway, and POU2F3, a transcription factor expressed by rare chemosensory cells called “tuft cells,” are both strongly expressed by some variant types of SCLC [24,25]. At least one of the four key molecules is expressed in SCLC cells, and Rudin et al. proposed a new model to categorize SCLC subtypes based on the expression of the following molecules: ASCL1 (SCLC-A), NEUROD1 (SCLC-N), YAP1 (SCLC-Y), and POU2F3 (SCLC-P). The accumulation of information on SCLC resulted in it being widely established as a heterogenous cancer. These four key molecules may contribute to the heterogeneity of SCLC tumors; namely, tumor growth differences, prognosis, and chemotherapy resistance; thus, these molecules have potential as new therapeutic targets [11]. Currently, it remains unclear whether IHC on ASCL1, NEUROD1, YAP1, and POU2F3 may be applied for the classification of SCLC molecular subtypes.

We initially examined the relationships among the four key molecules and their target molecules using the gene expression omnibus dataset of Asian SCLC patients [26]. This public dataset showed that some molecules exhibited a wide range of expression lesions in association with the four key molecules. Therefore, we investigated the expression of target molecules associated with the four key molecules in surgically resected SCLC samples. An integrated SCLC study that uses IHC to investigate the relationship between these molecules has not yet been conducted; therefore, we attempted to demonstrate the importance of IHC by evaluating the expression of these molecules. The present study aimed to establish the utility of IHC for identifying SCLC molecular subtypes and to assess the expression of these target molecules in resected SCLC samples. Differences in the expression patterns among SCLC molecular subtypes may be associated with SCLC characteristics, such as drug resistance and relapse. Therefore, in addition to gene expression levels, intracellular and intranuclear protein expression levels may be important factors contributing to the identification of SCLC characteristics among the molecular subtypes. Based on the potential use of these molecules as molecular targeted therapies for SCLC, IHC may facilitate the diagnosis of SCLC molecular subtypes and aid in the selection of appropriate individualized cancer treatments, which may be more effective than chemotherapy.

## 2. Materials and Methods

### 2.1. Cell Lines and Xenotransplantation

Four SCLC cell lines (H209, H82, H526, and SBC3) were used in the experiments. H209 cells were purchased from ATCC (Manassas, VA, USA), SBC3 cells were purchased from the Japan Collection of Research Bioresources Cell Bank (Osaka, Japan), and H82 and H56 cells were generous gifts from Dr. K. Hasegawa (Kyoto Pharmaceutical University, Kyoto, Japan). Cultured cells (1.0 × 10^6^ cells) were subcutaneously injected into Rag2(−/−); Jak3(−/−) mice (a generous gift from Prof. S. Okada, Kumamoto University). After 4 weeks, the subcutaneous tumors that grew were removed and fixed with 4% paraformaldehyde in a phosphate-buffered solution before being embedded in paraffin. Paraffin-embedded sections were stained with hematoxylin and eosin and immunostained for ASCL1, NEUROD1, YAP1, and POU2F3.

### 2.2. Western Blot (WB) Analysis

The SCLC cell lines were used for western blot (WB) analyses of the four key molecules. The primary antibodies used for WB analysis are listed in Table 1. The membrane was washed and incubated with a secondary antibody conjugated with horseradish peroxidase for 1 h, and the immune complex was visualized with a chemiluminescence substrate (Amersham Pharmacia Biotech, Buckinghamshire, UK).

### 2.3. Tissue Samples

Tissue samples of SCLC (*n* = 47) resected at the Department of Thoracic Surgery of Kumamoto University Hospital (Kumamoto, Japan) and the Department of Thoracic Surgery of National Minami-Kyushu Hospital (Kagoshima, Japan) were used in the present study. Samples were histologically diagnosed according to the WHO criteria [12]. Additional sections were used for IHC. The present study followed the guidelines of the Ethics Committee of Kumamoto University and National Minami-Kyushu Hospital on May 15, 2017 (Approval No. 342).

### 2.4. IHC

Formalin-fixed paraffin-embedded specimens were cut into 4-μm-thick sections and mounted onto MAS-GP–coated slides (Matsunami Glass Ind., Osaka, Japan). After deparaffinization and rehydration, sections were heated using an autoclave in 0.01 mol/L citrate buffer (pH 6.0, 7.0, or 9.0) for antigen retrieval. Antigen retrieval was not performed in POU2F3 staining. Sections were incubated with 0.3% H_2_O_2_ in absolute methanol for 30 min to block endogenous peroxidase activity. Sections were incubated with 5% skim milk for 20 min to block nonspecific binding and were then incubated with the primary antibody at 4 °C overnight or at room temperature for 70 min. This was followed by a 1-h incubation period with the secondary antibody (En Vision+ System-HRP-Labeled Polymer; Dako (Agilent, Carpinteria, CA, USA) and visualization with the Liquid DAB+ Substrate Chromogen System (Dako). All slides were counterstained with hematoxylin for 30 s before dehydration and mounting. The specificity of the immunolabeling of each antibody was examined using normal mouse IgG (Santa Cruz Biotechnology, Dallas TX, USA) and normal rabbit IgG (Santa Cruz Biotechnology), and no staining was observed.

### 2.5. IHC Analysis

Staining intensities and areas stained using IHC were scored according to a 4-tier system: 0 = no intensity or a staining area less than 10%; 1 = weakly positive intensity and 10–50% SCLC tumor cells stained; 2 = weakly positive intensity and > 50% SCLC tumor cells stained or strongly positive intensity and 10–50% SCLC tumor cells stained; and 4 = strongly positive intensity and > 50% SCLC tumor cells stained.

### 2.6. Statistical Analysis

All statistical analyses were performed with R for Windows (Version 4.0.2; R Foundation for Statistical Computing, Vienna, Austria). The Shapiro–Wilk test was used as the normality test. When the data did not follow a normal distribution, the Mann–Whitney test was used to test for significance. *p* values < 0.05 were considered to indicate a significant difference.

## 3. Results

### 3.1. Expression of Four Key Molecules in Representative SCLC Cell Lines

The expression of the key molecules in the representative SCLC cell lines was assessed. WB showed that H209 cells only expressed ASCL1. H82 cells strongly expressed NEUROD1 and weakly expressed POU2F3. H526 cells only expressed POU2F3, and SBC3 cells only expressed YAP1 (Figure 1A). Xenotransplanted tumors originating from the four SCLC cell lines exhibited SCLC histological features and positive staining for ASCL1 in H209 cell-derived tumors, NEUROD1 in H82 cell-derived tumors, POU2F3 in H526 cell-derived tumors, and YAP1 in SBC3 cell-derived tumors (Figure 1B).

### 3.2. ASCL1 Exhibited the Highest Positive Rate among the Four Key Molecules in SCLC

The IHC results of all molecules are summarized in Table 2. Among the four key molecules, ASCL1 showed the highest positive rate (72.3%) in SCLC samples followed by POU2F3 (38.4%), YAP1 (14.9%), and NEUROD1 (10.6%). ASCL1 showed moderate to strong expression in contrast to the weak expression of NEUROD1, YAP1, and POU2F3. The neuroendocrine markers CGA, SYP, NCAM, and INSM1 showed positive rates of approximately 75–90% in resected SCLC samples. Among the proteins tested, E2F7 showed the highest positive rate (100.0%), while MYC showed the lowest (2.1%).

### 3.3. Four Key Molecules Were Simultaneously Expressed in SCLC

ASCL1 exhibited coordinated expression with the three other key molecules in nearly 50% of ASCL1-positive cases. POU2F3 showed a higher co-expression rate (32.4%) than NEUROD1 (8.8%) and YAP 1 (11.8%) in ASCL1-positive cases (Table 3). The expression levels of these molecules in ASCL1-positive cases were low (NEUROD1: 100.0%, YAP1: 75.0%, POU2F3: 63.6%). Simultaneous expression of the four key molecules was frequently observed in SCLC samples (Table 4). The most common expression pattern was ASCL1 alone (38.8%) followed by double positivity for ASCL1 and POU2F3 (18.4%), POU2F3 alone (10.2%), double positivity for ASCL1 and YAP1 (6.1%), YAP1 alone (4.1%), and double positivity for ASCL1 and NEUROD1 (2.0%). Triple-positive (ASCL1, NEUROD1, and POU2F3) or all-positive cases were also observed in the present study. Additionally, the expression areas of each of the four key molecules were frequently shared. Similar staining patterns were observed among ASCL1, NEUROD1, YAP1, and POU2F3 (Figure 2), particularly for NEUROD1 and YAP1, NEUROD1 and POU2F3, and YAP1 and POU2F3 (Figure 3).

### 3.4. Candidate Target Molecule Expression in ASCL1-Positive and ASCL1-Negative Samples

To assess the expression of candidate target molecules in SCLC by IHC, we attempted to stain resected SCLC samples for 26 target molecules other than ASCL1, NEUROD1, YAP1, and POU2F3. Resected SCLC samples were grouped by a positive or negative result for ASCL1, NEUROD1, YAP1, and POU2F3, and differences in the expression of the candidate target molecules were compared between the ASCL1-positive and ASCL1-negative groups, NEUROD1-positive and NEUROD1-negative groups, YAP1-positive and YAP1-negative groups, and POU2F3-positive and POU2F3-negative groups. The candidate target molecule expression associated with ASCL1 expression is summarized in Table 5. Significant differences were observed in the expression of CGA, SYP, NCAM, INSM1, NOTCH3, SOX2, VIMENTIN, TTF1, FGFR1, EZH2, TEAD1, P53, and DLL3 between the ASCL1-positive and ASCL1-negative groups (*p* < 0.05). The expression of neuroendocrine markers (CGA, SYP, NCAM, and INSM1) and SOX2, TTF1, EZH2, TEAD1, and DLL3 was slightly stronger in the ASCL1-positive group than in the ASCL1-negative group, whereas the expression of NOTCH3, FGFR1, VIMENTIN, and P53 was slightly weaker.

### 3.5. Candidate Target Molecule Expression in NEUROD1-Positive and NEUROD1-Negative Samples

The candidate target molecule expression associated with NEUROD1 expression is summarized in Table 6. Significant differences were detected in the expression of E-CADHERIN, E2F1, RB1, and P53 between the NEUROD1-positive and NEUROD1-negative groups (*p* < 0.05). The expression of E-cadherin, E2F1, RB1, and P53 was slightly stronger in the NEUROD1-positive group than in the NEUROD1-negative group.

### 3.6. Candidate Target Molecule Expression in YAP1-Positive and YAP1-Negative Samples

The candidate target molecule expression associated with YAP1 expression is summarized in Table 7. Significant differences were observed in the expression of TTF1, E2F1, and P53 between the YAP1-positive and YAP1-negative groups (*p* < 0.05). The expression of P53 was slightly stronger in the YAP1-positive group than in the YAP1-negative group, whereas the expression of TTF1 and E2F7 was slightly weaker.

### 3.7. Candidate Target Molecule Expression in POU2F3-Positive and POU2F3-Negative Samples

The candidate target molecule expression associated with POU2F3 expression is summarized in Table 8. Significant differences were observed in the expression of INSM1 and HES1 between the POU2F3-positive and POU2F3-negative groups (*p* < 0.05). The expression of INSM1 and HES1 was slightly stronger in the POU2F3-positive group than in the POU2F3-negative group.

## 4. Discussion

Approximately 70% of SCLC specimens show ASCL1 expression, and approximately 15% of SCLC specimens show NEUROD1 expression [22,27]. YAP1 expression is very rare (2%) in SCLC [28], while POU2F3 expression has been detected in 12% of SCLC samples using tissue microarrays [25]. In the present study, ASCL1 was expressed in approximately 70% of the resected SCLC samples, NEUROD1 in approximately 10%, YAP1 in approximately 15%, and POU2F3 in approximately 40%. The proportions of ASCL1- and NEUROD1-positive cases in the present study were consistent with previous findings [22,27], whereas those of YAP1- and POU2F3-positive cases were higher than previously reported [25,28]. This increase found in the present study may be attributed to differences in the IHC method, the quality of sections, and the antibodies used. Furthermore, the expression levels of YAP1 and POU2F3 in SCLC cells may change more easily than those of ASCL1 and NEUROD1.

The present results showed that SCLC did not always express one of the key molecules (Table 3 and Table 4). Among the resected SCLC samples, 50% expressed one specific key molecule, while several co-expression patterns were observed in the other 50%. In some cases, the combinations of ASCL1 and POU2F3 or NEUROD1 and POU2F3 were doubly positive, and it was not possible to group cases based on SCLC molecular subtypes using IHC. Therefore, the four key molecules in SCLC were shown to have several expression patterns, and SCLC samples may not all be matched to molecular subtypes. A more detailed method for IHC needs to be developed to more accurately differentiate SCLC molecular subtypes using the microscopic differences in protein expression. 

Regarding positive areas, Figure 2 shows that the ASCL1-positive regions in SCLC specimens correspond to NEUROD1-, YAP1-, or POU2F3-positive regions. Figure 3 also shows that the NEUROD1-positive regions in SCLC samples correspond to YAP1- and POU2F3-positive regions, as YAP1-positive regions correspond to POU2F3-positive regions. These results suggest that the four key molecules are coordinately and simultaneously expressed in SCLC cells. The co-expression of the four key molecules in SCLC samples suggests an interaction among these molecules. A recent study reported that SCLC-expressing ASCL1 shifted to the expression of NEUROD1 and YAP1 following the activation of MYC [29]. The expression of NEUROD1 and YAP1 in ASCL1-positive regions indicates a contemporary shift from ASCL1 to other key molecules, and these changes may be closely associated with the transition of SCLC molecular subtypes.

Significant differences were observed in the expression of several proteins between the ASCL1-positive and ASCL1-negative groups. ASCL1-positive SCLC may be classified as the classic subtype, which has the typical morphology with neuroendocrine features. Neuroendocrine features were more prominent in the ASCL1-positive group than in the ASCL1-negative group. In the ASCL1-positive group, INSM1 exhibited the most sensitive expression compared with CGA, SYP, and NCAM. The POU2F3-positive group appeared to correspond to the non-neuroendocrine group of SCLC based on the significant decrease observed in INSM1 expression. INSM1 is a highly sensitive and specific marker of neuroendocrine differentiation [14,15], and a strong correlation between the expression of ASCL1 and INSM1 in neuroendocrine differentiation has been reported [30,31,32]. NEUROD1-positive SCLC is another neuroendocrine subtype of SCLC; however, no significant differences in the expression of general neuroendocrine markers between the NEUROD1-positive and NEUROD1-negative groups were noted. Therefore, NEUROD1 protein expression does not appear to play an important role in the expression of neuroendocrine features in SCLC. Similarly, no or slight differences were observed in the expression of general neuroendocrine markers between the YAP1- and POU2F3-positive, non-neuroendocrine, and non-expression groups.

Significant differences were observed in the expression of some proteins, excluding neuroendocrine markers, between the ASCL1-positive and ASCL1-negative groups. The expression of SOX2, TTF1, EZH2, DLL3, and TEAD1 was significantly upregulated in the ASCL1-positive group. SOX2 is a transcriptional regulator of neuroendocrine differentiation [31,33], and TTF1 is expressed in club, alveolar type Ⅱ, and bronchial neuroendocrine cells [34]. A previous study reported that the expression of EZH2 promoted the progression of SCLC by suppressing the TGF-β-SMAD-ASCL1 pathway [35], while the expression of DLL3 was positively correlated with that of ASCL1 in SCLC [36]. These molecules, excluding TEAD1, may be expressed in ASCL1-positive SCLC, which is supported by the results of this study. TEAD1 may contribute to the promotion of tumorigenesis via the YAP/TAZ signaling pathway; however, few studies have examined the relationship between ASCL1 and TEAD1 [37]. In contrast, the expression levels of NOTCH3, FGFR1, VIMENTIN, and P53 were significantly decreased in the ASCL1-positive group. Notch signaling suppresses neuroendocrine differentiation and is inactivated in most SCLC cases; therefore, the Notch family may be inactive in the classic SCLC subtype [38,39,40,41]. FGFR1 and its ligands have therapeutic potential as new treatments for SCLC [42]. FGFR1 overexpression in SCLC may lead to new FGFR1 inhibitor therapy, but the FGFR1 expression in the ASCL1-positive group was decreased. Thus, these findings suggest that a new FGFR1 inhibitor therapy may not be useful for some patients with ASCL1-positive SCLC.

The expression of some proteins significantly differed between the ASCL1-positive group and the NEUROD1-, YAP1-, and POU2F3-positive groups. Nevertheless, the immunoreactivity of RB1 and P53 was significantly stronger in the NEUROD1-positive group than in the NEUROD1-negative group. Similarly, the immunoreactivity of P53 was significantly stronger in the YAP1-positive group than in the YAP1-negative group. RB1 and P53 tumor suppressor gene alterations and high mutational burden frequently occur in SCLC [4,18,26,38,43]. The inactivation of these tumor suppressor genes in SCLC is common [18] and frequently occurs in ASCL1-positive SCLC. It currently remains unclear why areas of RB1 and P53 immunoreactivity were observed in the NEUROD1- and YAP1-positive groups. The mutation regions in these suppressor oncogenes do not always lose antibody-recognition sites, and their positive immunoreactivity may have occurred in the present study.

Additionally, significant differences were observed in the expression of E2F1 and E2F7 in the NEUROD1- and YAP1-positive groups. E2F1 is a member of the E2F family and promotes epithelial-mesenchymal transition (EMT), which is associated with SCLC invasion and metastasis [44]. The positive rate of E2F1 expression in all SCLC samples was lower than other protein expression rates but was higher in the NEUROD1-positive group than in the NEUROD1-negative group. E2F7 may be a transcriptional repressor of cell proliferation and exhibited the highest positive rate (100%) in all SCLC samples. It also plays a role in tumor development.

No significant differences were observed in the expression of MYC among any of the groups. The MYC family comprises MYC, L-MYC, and N-MYC, and MYC may be amplified in approximately 20% of SCLC cases [45]. A recent study reported that MYC promotes a temporal shift in SCLC molecular subtypes from ASCL1 → NEUROD1 → YAP1 [29]. Notably, MYC may play an important role in changes in SCLC molecular subtypes that are associated with chemotherapy resistance and new treatment development. We expected the expression of MYC to increase in the NEUROD1- and YAP1-positive groups and to decrease in the ASCL1-positive group. However, no or only slight differences were observed in the expression of MYC in our SCLC samples.

## 5. Conclusions

The positive expression rates of ASCL1 and NEUROD1 in our SCLC samples were similar to those found in previous reports [22,27]. The expression of the target molecules in the ASCL1-positive group was consistent with previous findings [31,33,34,35,36,37,38,39,40,41,42]. As half of the candidate ASCL1-target molecules selected from the gene expression omnibus dataset showed significance in the IHC study of the ASCL1-positive samples, IHC could be a useful method for detecting candidate target molecules for the four key molecules. We found that the four key molecules are coordinately and simultaneously expressed in SCLC cells. However, the positive rates of YAP1 and POU2F3 in our SCLC samples were not consistent with those previously reported [25,28]. Moreover, the majority of the candidate target molecules examined were not significantly expressed in NEUROD1-, YAP1- and POU2F3-positive samples. However, further improvements in the classification of SCLC by IHC are expected, and IHC could become an essential method for the determination of the molecular subtypes of SCLC.

## Figures and Tables

**Figure 1 diagnostics-10-00949-f001:**
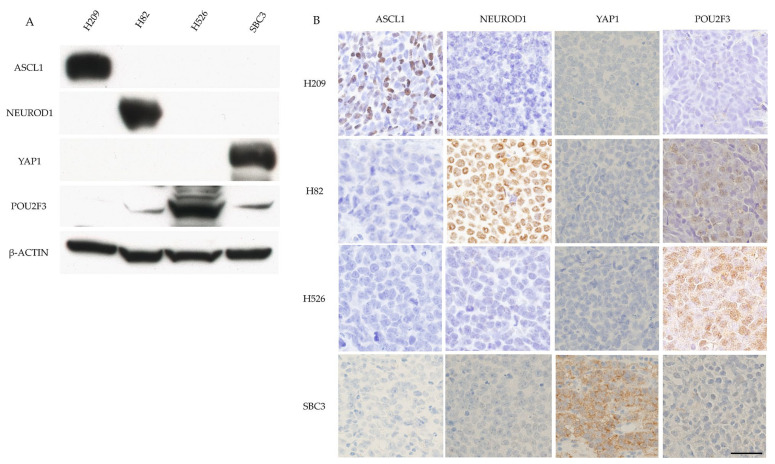
Example of the positive control of four key molecules by WB and IHC in SCLC samples. (**A**) Example of the positive control of the four key molecules by WB in SCLC cell lines. ASCL1 was strongly expressed in H209 cells. NeuroD1 was strongly expressed in H82 cells. Pou2F3 was strongly expressed in H526 cells and weakly expressed in H82 and SBC3 cells. YAP1 was strongly expressed in SBC3 cells. β-actin served as an internal control. (**B**) Example of the positive control of the four key molecules in xenotransplanted tumor tissues from the four cell lines in mice by IHC. ASCL1 staining was found in tumor cell nuclei of H209 cells. NeuroD1 staining was found in the tumor cell nuclei of H82 cells. Pou2F3 staining with a diffuse cytoplasmic pattern was found in H82, H526, and SBC3 cells. The Pou2F3 staining intensity was weak in H82 and SBC3 cells and strong in H526 cells. YAP1 was stained with a membranous pattern in SBC3 cells. Scale bar = 50 μm.

**Figure 2 diagnostics-10-00949-f002:**
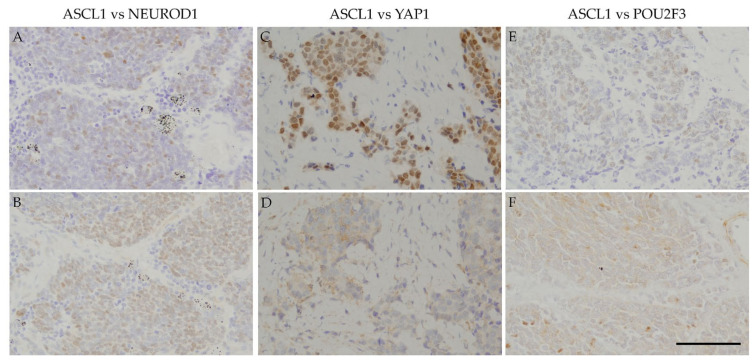
IHC of the co-expression of ASCL1 and three key molecules in SCLC samples: Serial section immunostaining for ASCL1 (A) and NEUROD1 (B). SCLC cell nuclei were positive for ASCL1 (**A**) and NEUROD1 (**B**). Serial section immunostaining for ASCL1 (**C**) and YAP1 (**D**). SCLC cell nuclei were positive for ASCL1 (**C**), and the cytoplasm was positive for YAP1 (**D**). SCLC cell nuclei were positive for ASCL1 (**E**), and the cytoplasm was positive for POU2F3 (**F**). Scale bar = 100 μm.

**Figure 3 diagnostics-10-00949-f003:**
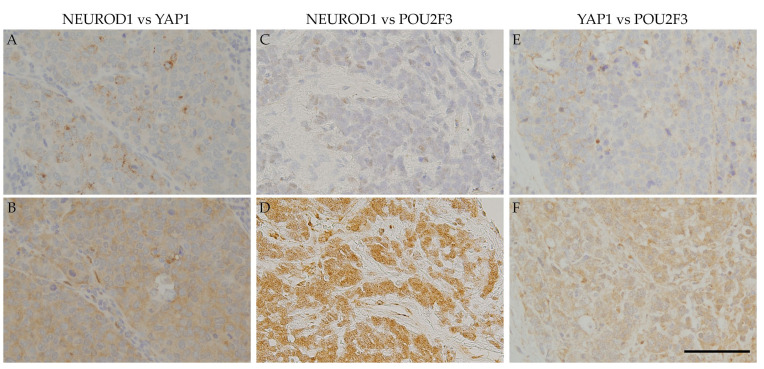
IHC of the co-expression of key molecules other than ASCL1 in SCLC samples. Serial section immunostaining for NEUROD1 (**A**) and YAP1 (**B**): SCLC cell nuclei were positive for NEUROD1 (**A**), and the cytoplasm was positive for YAP1 (**B**). Serial section immunostaining for NEUROD1 (**C**) and POU2F31 (**D**): SCLC cell nuclei were positive for NEUROD1 (**C**), and the cytoplasm was positive for POU2F3 (**D**). The cytoplasm of SCLC cells was positive for YAP1 (**E**) and POU2F3 (**F**). Scale bar = 100 μm.

**Table 1 diagnostics-10-00949-t001:** List and dilution of antibodies for IHC and the WB analysis

Antibody	Manufacturer (Catalog no.)	IHC	WB
ASCL1	Abcam (ab213151)	1/300	1/1000
NEUROD1	Novus (NBP1-88661)	1/100	1/1000
YAP1	Santa Cruz (sc-376830)	1/80	1/1000
POU2F3	LifeSpan BioSciences (LS-B12579)	1/100	1/1000
CGA	Santa Cruz (sc-13090)	1/100	
SYP	Leica (NCL-SYNP-299)	1/50	
NCAM	Leica (NCL-CD56-1B6)	1/50	
INSM1	Santa Cruz (sc-271408)	1/200	
NOTCH1	Cell Signaling (#3268)	1/200	
NOTCH2	Cell Signaling (#4530)	1/200	
NOTCH3	Abcam (ab23426)	1/100	
SOX2	Santa Cruz (sc-365823)	1/400	
SOX9	Millipore (AB5603)	1/1000	
E-CADHERIN	GeneTex (GTX100443)	1/100	
VIMENTIN	Santa Cruz (sc-6260)	1/100	
TTF1	Santa Cruz (sc-13040)	1/100	
FGFR1	Cell Signaling (#9740)	1/50	
MYC	Cell Signaling (#5605)	1/50	
E2F1	Santa Cruz (sc-251)	1/200	
E2F7	Thermo Fisher (PA550495)	1/200	
BCL2	DAKO (MO887)	1/50	
PDL1	Abcam (ab228415)	1/500	
EZH2	Cell Signaling (#5246)	1/50	
TEAD1	Cell Signaling (#12292)	1/70	
RB1	Novus (NB100-82177)	1/100	
P53	DAKO (DO-7)	1/100	
P130	Abcam (ab6545)	1/300	
DLL3	Cell Signaling (#71804)	1/50	
HES1	LifeSpan (LS-C337570)	1/50	
REST	GeneTex(GTX37363)	1/400	

IHC, immunohistochemistry; WB, Western blot.

**Table 2 diagnostics-10-00949-t002:** Immunohistochemical results in SCLC.

Antibody	Positive Rate (%)		Expression Level	
		Low	Medium	High
ASCL1	72.3	17.6	67.6	14.7
NEUROD1	10.6	80.0	20.0	0
YAP1	14.9	57.1	42.9	0
POU2F3	38.4	66.7	20.0	13.3
CGA	74.5	34.2	28.6	37.1
SYP	87.0	7.5	35.0	57.5
NCAM	91.3	14.3	42.9	42.9
INSM1	83.0	12.8	33.3	53.8
NOTCH1	17.4	75.0	12.5	12.5
NOTCH2	53.2	32.0	68.0	0
NOTCH3	66.0	19.4	51.6	29.0
SOX2	89.4	33.3	45.2	21.4
SOX9	89.4	26.2	35.7	38.1
E-CADHERIN	80.9	39.5	42.1	18.4
VIMENTIN	6.4	0	100	0
TTF1	74.5	2.9	42.9	54.3
FGFR1	19.6	55.6	22.2	22.2
MYC	2.1	100	0	0
E2F1	11.1	60.0	40.0	0
E2F7	100	4.3	27.7	68.1
BCL2	78.7	24.5	40.5	35.1
PDL1	40.4	36.8	36.8	26.3
EZH2	85.1	30.0	60.0	10.0
TEAD1	80.0	47.2	47.2	5.6
RB1	42.2	68.4	31.6	0
P53	84.8	12.8	43.6	43.6
P130	57.4	18.5	77.8	7.4
DLL3	37.0	52.9	29.4	17.6
HES1	51.1	8.3	83.8	8.3
REST	19.1	11.1	88.9	0

**Table 3 diagnostics-10-00949-t003:** Co-expression of NEUROD1, YAP1, and POU2F3 in the ASCL1-positive SCLC group

Antibody	Positive Rate (%)		Expression Level	
		Low	Medium	High
NEUROD1	8.8	100	0	14.7
YAP1	11.8	75.0	25.0	0
POU2F3	32.4	63.6	18.2	18.2

**Table 4 diagnostics-10-00949-t004:** Combined expression of ASCL1, NEUROD1, YAP1, and POU2F3

	Expression Pattern	Percentage (%)
Single-positive	ASCL1 only	38.8
	POU2F3 only	10.2
	YAP1 only	4.1
Double-positive	ASCL1, POU2F3	18.4
	ASCL1, YAP1	6.1
	ASCL1, NEUROD1	2.0
	NEUROD1, YAP1	2.0
	NEUROD1, POU2F3	2.0
Triple-positive	ASCL1, NEUROD1, POU2F3	2.0
All-positive		2.0
All-negative		12.2

**Table 5 diagnostics-10-00949-t005:** Candidate target molecular expression in ASCL1-positive and ASCL1-negative samples.

Antibody		ASCL1-Positive			ASCL1-Negative		*p*-Value
	Low	Medium	High	Low	Medium	High	
CGA	20.6	20.6	38.2	30.8	30.8	0	0.02235
SYP	8.8	35.3	55.9	0	16.7	33.3	0.00731
NCAM	5.9	35.3	47.1	33.3	33.3	16.7	0.00563
INSM1	5.9	41.2	47.1	23.1	7.7	7.7	0.000001108
NOTCH3	17.6	32.4	8.8	0	38.5	46.2	0.00628
SOX2	29.4	38.2	26.5	30.8	46.2	0	0.02425
VIMENTIN	0	0	0	0	23.1	0	0.00631
TTF1	2.9	41.2	47.1	0	7.7	23.1	0.00131
FGFR1	5.9	2.9	2.9	25.0	8.3	8.3	0.04615
EZH2	23.5	55.9	11.8	30.8	38.5	0	0.01039
TEAD1	31.3	50.0	6.3	53.8	7.7	0	0.00136
P53	12.1	39.4	27.3	7.7	30.8	61.5	0.00623
DLL3	27.3	15.2	6.1	0	0	7.7	0.01203

**Table 6 diagnostics-10-00949-t006:** Candidate target molecular expression in NEUROD1-positive and NEUROD1-negative samples.

Antibody		NEUROD1-Positive			NEUROD1-Negative		*p*-Value
	Low	Medium	High	Low	Medium	High	
E-cadherin	0	40.0	60.0	35.7	33.3	9.5	0.00535
E2F1	20.0	20.0	0	5.0	2.5	33.3	0.03138
RB1	40.0	60.0	0	26.8	7.3	16.7	0.00123
P53	0	0	100	12.2	41.5	29.3	0.00854

**Table 7 diagnostics-10-00949-t007:** Candidate target molecular expression in YAP1-positive and YAP1-negative samples.

Antibody		YAP1-Positive			YAP1-Negative		*p*-Value
	Low	Medium	High	Low	Medium	High	
TTF1	0	28.6	0	2.5	32.5	47.5	0.00434
E2F7	14.3	57.1	28.6	2.5	22.5	75.0	0.01398
P53	0	14.3	85.7	12.8	41.0	28.2	0.00826

**Table 8 diagnostics-10-00949-t008:** Candidate target molecular expression in POU2F3-positive and POU2F3-negative samples.

Antibody		POU2F3-Positive			POU2F3-Negative		*p*-Value
	Low	Medium	High	Low	Medium	High	
INSM1	13.3	13.3	66.7	12.5	33.3	25.0	0.044
HES1	6.7	53.3	6.7	4.2	29.2	0	0.04094

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
