# Peer review of "Integrated Immunohistochemical Study on Small-Cell Carcinoma of the Lung Focusing on Transcription and Co-Transcription Factors"

_diagnostics, 2020, doi:10.3390/diagnostics10110949_

Round 1
Reviewer 1 Report
1) The study is pleasant to read. The paper is overall well-written, but occasional minor typographic and grammatical errors are present. Recommend further proof-reading and editorial help from English speaking person prior to final publication. 2) The “abstract” should be revised because it does not align with the text. 3) The IHC figures (1, 2, and 3) are without scale bars. 4) The IHC figures appear at very low resolutions. 5) You can consult and add the following papers: PMID: 32987854; PMID: 27138756; PMID: 29544351.
Author Response
Thank you very much for inviting us to submit a revised draft of our manuscript entitled, “Integrated immunohistochemical study on small cell carcinoma of the lung focusing on transcription and co-transcription factors” to Diagnostics. We also appreciate the time and effort you and each of the reviewers have dedicated to providing insightful feedback on ways to strengthen our paper. Thus, it is with great pleasure that we resubmit our article for further consideration. We have incorporated changes that reflect the detailed suggestions you have graciously provided. We also hope that our edits and the responses we provide below satisfactorily address all the issues and concerns you and the reviewers have noted
To facilitate your review of our revisions, the following is a point-by-point response to the questions and comments delivered in your letter dated 9th October 2020.
Reviewer: 1
Comments and Suggestions for Authors
1) The study is pleasant to read. The paper is overall well-written, but occasional minor typographic and grammatical errors are present. Recommend further proof-reading and editorial help from English speaking person prior to final publication. 2) The “abstract” should be revised because it does not align with the text. 3) The IHC figures (1, 2, and 3) are without scale bars. 4) The IHC figures appear at very low resolutions. 5) You can consult and add the following papers: PMID: 32987854; PMID: 27138756; PMID: 29544351.
Point 1: The study is pleasant to read. The paper is overall well-written, but occasional minor typographic and grammatical errors are present. Recommend further proof-reading and editorial help from English speaking person prior to final publication.
Response 1: Thank you for providing these insights. I'm very sorry I was not familiar with English sentences. We requested native speakers of English to proofread our manuscript.
Point 2: The “abstract” should be revised because it does not align with the text.
Response 2: Thank you for your suggestion. We reconsidered the results and conclusion of the paper, and we have rewritten and revised the abstract (p1, lines;30-35).
“ASCL1 showed the highest positive rate in SCLC samples, and significant differences were observed in the expression levels of some target molecules between the ASCL1-positive and ASCL1-negative groups. Furthermore, the four key molecules were coordinately and simultaneously expressed in SCLC cells. An IHC study of ASCL1-positive samples showed many candidate SCLC target molecules, and IHC could become an essential method for determining SCLC molecular subtypes.”
Point 3: The IHC figures (1, 2, and 3) are without scale bars.
Response 3: We agree with you. We have added a scale bar (figure1:50µm, figure2 and 3:100µm) in IHC figures.
Point 4: The IHC figures appear at very low resolutions.
Response 4: We agree with your assessment. We have changed the resolution of the IHC figures. The resolution of new IHC figures is 350dpi.
Point 5: You can consult and add the following papers: PMID: 32987854; PMID: 27138756; PMID: 29544351.
Response 5: Thank you for your constructive comments. We have referred to the papers you proposed (PMID: 32987854; PMID: 27138756; PMID: 29544351.) and have added these papers to reference. The added texts are (p2, lines;49-50) “therefore, it is necessary to identify and develop novel therapeutic approaches to treat SCLC [8(PMID: 27138756), 9(PMID: 29544351)].” and (p10, lines;318-319) “RB1 and P53 tumor suppressor gene alterations and high mutational burden frequently occur in SCLC [4,18,26,38,43(PMID: 32987854)].”
Again, thank you for giving us the opportunity to strengthen our manuscript with your valuable comments and queries. We have worked hard to incorporate your feedback and hope that these revisions persuade you to accept our submission.
Sincerely,
Younosuke Sato, M.D.
Graduate School of Medical Sciences Department of Pathology and Experimental Medicine, Kumamoto University, 1-1-1 Honjo, Chuo-ku, Kumamoto, Kumamoto 860-8556, Japan
nosuke2014@gmail.com
TEL +81-96-373-5089, FAX +81-96-373-5089

Reviewer 2 Report
In the conclusions chapter, I do not understand why the analysis of ASCL1 protein expression (alone?) has the potential to discriminate between ASCL1-, NEUROD1-, YAP1-, and POU2F3-positive SCLC.
Author Response
Thank you very much for inviting us to submit a revised draft of our manuscript entitled, “Integrated immunohistochemical study on small cell carcinoma of the lung focusing on transcription and co-transcription factors” to Diagnostics. We also appreciate the time and effort you and each of the reviewers have dedicated to providing insightful feedback on ways to strengthen our paper. Thus, it is with great pleasure that we resubmit our article for further consideration. We have incorporated changes that reflect the detailed suggestions you have graciously provided. We also hope that our edits and the responses we provide below satisfactorily address all the issues and concerns you and the reviewers have noted.
To facilitate your review of our revisions, the following is a point-by-point response to the questions and comments delivered in your letter dated 9th October 2020.
Reviewer: 2
Comments and Suggestions for Authors
In the conclusions chapter, I do not understand why the analysis of ASCL1 protein expression (alone?) has the potential to discriminate between ASCL1-, NEUROD1-, YAP1-, and POU2F3-positive SCLC
Response: Thank you for your suggestion. I am sorry that the conclusion chapter was unclear in the original manuscript. We reconsidered our manuscript, and the content was insufficient to demonstrate the potential to discriminate ASCL1-positive SCLC from the NeuroD1, YAP1, POU2F3-positive SCLC. We have rewritten and revised the conclusion text (p11, lines;341-352) to align with the results of our manuscript.
- Conclusions
“The positive expression rates of ASCL1 and NEUROD1 in our SCLC samples were similar to those found in previous reports [22,27]. The expression of the target molecules in the ASCL1-positive group was consistent with previous findings [31,33–42]. As half of the candidate ASCL1-target molecules selected from the gene expression omnibus dataset showed significance in the IHC study of the ASCL1-positive samples, IHC could be a useful method for detecting candidate target molecules for the four key molecules. We found that the four key molecules are coordinately and simultaneously expressed in SCLC cells. However, the positive rates of YAP1 and POU2F3 in our SCLC samples were not consistent with those previously reported [25,28]. Moreover, the majority of the candidate target molecules examined were not significantly expressed in NEUROD1-, YAP1- and POU2F3-positive samples. However, further improvements in the classification of SCLC by IHC are expected, and IHC could become an essential method for the determination of the molecular subtypes of SCLC.”
Again, thank you for giving us the opportunity to strengthen our manuscript with your valuable comments and queries. We have worked hard to incorporate your feedback and hope that these revisions persuade you to accept our submission.
Sincerely,
Younosuke Sato, M.D.
Graduate School of Medical Sciences Department of Pathology and Experimental Medicine, Kumamoto University, 1-1-1 Honjo, Chuo-ku, Kumamoto, Kumamoto 860-8556, Japan
nosuke2014@gmail.com
TEL +81-96-373-5089, FAX +81-96-373-5089
